# Reduced Number and Immune Dysfunction of CD4+ T Cells in Obesity Accelerate Colorectal Cancer Progression

**DOI:** 10.3390/cells12010086

**Published:** 2022-12-25

**Authors:** Kota Yamada, Masafumi Saito, Masayuki Ando, Tomoki Abe, Tomosuke Mukoyama, Kyosuke Agawa, Akihiro Watanabe, Shiki Takamura, Mitsugu Fujita, Naoki Urakawa, Hiroshi Hasegawa, Shingo Kanaji, Takeru Matsuda, Taro Oshikiri, Yoshihiro Kakeji, Kimihiro Yamashita

**Affiliations:** 1Division of Gastrointestinal Surgery, Department of Surgery, Graduate School of Medicine, Kobe University, Kobe 650-0017, Japan; 2Department of Disaster and Emergency and Critical Care Medicine, Graduate School of Medicine, Kobe University, 7-5-2, Kusunoki-cho, Chuo-ku, Kobe 650-0017, Japan; 3Department of Immunology, Kindai University Faculty of Medicine, 377-2 Ono-higashi, Osakasayama 589-0014, Japan; 4Center for Medical Education and Clinical Training, Kindai University Faculty of Medicine, 377-2 Onohigashi, Osaka 589-0014, Japan

**Keywords:** obesity, colorectal cancer, high-fat diet, CD4+ T cell, tumor immune microenvironment

## Abstract

Obesity, a known risk factor for various types of cancer, reduces the number and function of cytotoxic immune cells in the tumor immune microenvironment (TIME). However, the impact of obesity on CD4+ T cells remains unclear. Therefore, this study aimed to clarify the impact of obesity on CD4+ T cells in the TIME. A tumor-bearing obese mouse model was established by feeding with 45% high-fat diet (HFD), followed by inoculation with a colon cancer cell line MC38. Tumor growth was significantly accelerated compared to that in mice fed a control diet. Tumor CD4+ T cells showed a significant reduction in number and an increased expression of programmed death-1 (PD-1), and decreased CD107a expression and cytokine such as IFN-γ and TNF-α production, indicating dysfunction. We further established CD4+ T cell-depleted HFD-fed model mice, which showed reduced tumor infiltration, increased PD-1 expression in CD8+ T cells, and obesity-induced acceleration of tumor growth in a CD4+ T cell-dependent manner. These findings suggest that the reduced number and dysfunction of CD4+ T cells due to obesity led to a decreased anti-tumor response of both CD4+ and CD8+ T cells to ultimately accelerate the progression of colorectal cancer. Our findings may elucidate the pathogenesis for poor outcomes of colorectal cancer associated with obesity.

## 1. Introduction

Colorectal cancer (CRC) is the third most common cancer and the second most common cause of cancer-related deaths worldwide [1]. The number of new cases and deaths from CRC is increasing rapidly [2], necessitating new strategies for its prevention and treatment. Recent epidemiological studies revealed that obesity is a risk factor for various cancers, including CRC [3,4,5]. The global prevalence of obesity nearly tripled between 1975 and 2016, with 39% of adults being overweight (body mass index (BMI) ≥ 25 kg/m^2^) and 13% being obese (BMI ≥ 30 kg/m^2^) worldwide [6]. The latest research suggests that the risk of CRC correlates with increasing BMI [7]. Furthermore, it is estimated that the number of patients with CRC will further increase and the number of new cases and deaths will exceed 2.2 million and 1.1 million, respectively, by 2030 [2].

Previous studies have shown that obesity-associated cancers are associated with overproduction of hormones (estrogen, leptin, and insulin-like growth factor), induction of reactive oxygen species (ROS) production by free fatty acids, and changes in the gut microbiome [8,9]. In addition, one mechanism attributed to the increased risk of carcinogenesis and accelerated tumor growth in obesity is that obesity promotes immune cell dysfunction in the tumor immune microenvironment (TIME). Specifically, the increase of fatty acid levels due to obesity causes metabolic reprogramming and decreased cytotoxic activity of natural killer (NK) cells, which accelerate tumor progression [10]. Similarly, obesity has been reported to induce reprogramming of tumor cells to lipid metabolism, inhibit cytotoxic activity of CD8+ T cells [11], and increase immune senescence, tumor progression, and PD-1-mediated CD8+ T cell dysfunction [12]. Thus, the relationship between obesity and TIME has been revealed in recent years [13].

In the TIME, CD4+ T cells are traditionally considered as “helpers” of CD8+ T cells; however, recent studies have shown that a cluster of CD4+ T cells have cytotoxic activity and anti-tumor effects like CD8+ T cells [14,15]. Although obesity has been shown to be biased toward CD4+ T cell differentiation and directly affect trafficking ability [16], little is known about the impacts of obesity on the anti-tumor activity of CD4+ T cells [17]. Accordingly, this study aimed to clarify how obesity affects the immunosurveillance function and anti-tumor activity of CD4+ T cells in CRC using mouse models.

## 2. Materials and Methods

### 2.1. Animals and Study Design

Male C57BL/6J mice were obtained from CLEA Japan, Inc. (Tokyo, Japan), and all animal experiments were conducted in the Department of Laboratory Animal Science at Kobe University. All mice were maintained in a humidified, quiet room on a 12 h dark-light cycle at 22 ± 1 ℃, free of pathogens, fed a specified diet, and had free access to water. The mice were kept in plastic cages (25 × 15 × 17 cm) with five animals per cage. The mice were allowed to acclimatize for one week before the experiments, and the average weight of the mice at the beginning of the experiment was 21.5 g. Once a day, the health and behavior of the mice were observed. Aspirin was used for analgesia, and 4% isoflurane inhalation was used for anesthesia.

All mice were fed either a control diet (CD) containing 10% calories from fat (D12450B, Research Diets) or a high-fat diet (HFD) containing 45% calories from fat (D12451B, Research Diets) at 7 weeks of age. From a viewpoint of animal welfare, the humane endpoints were set as follows: difficulty in feeding and fluid intake, agonizing symptoms (self-injurious behavior, abnormal posture, breathing problems, and crying); and marked abdominal distention, tumor necrosis, ulceration, or tumor infection. Mice judged to have reached the humane endpoint were euthanized via cervical dislocation under anesthesia. In this study, a total of 105 C57BL/6J mice were used.

### 2.2. Cell Line

C57BL/6J mouse-derived CRC cell line MC38 was used for model establishment in this study. This cell line was originally established by Dr. F. James Primus at the Beckman Research Institute (Duarte, CA, USA) [18] and was kindly provided by Dr. Toshiyasu Ojima of Wakayama Medical University (Wakayama, Japan) [19]. MC38 cells were maintained in RPMI-1640 medium supplemented with 10% (*v*/*v*) heat-inactivated fetal bovine serum (Sigma-Aldrich, St Louis, MO, USA) at 37 °C in a 5% CO_2_ atmosphere. The cultured cells were confirmed to be negative for mycoplasma and viral contamination.

### 2.3. Induction of CRC in Obese Mice

Seven-week-old male C57BL/6J mice were randomly divided into two groups and fed the CD or HFD (n = 20 per each group) for 9 weeks. Blood samples were collected from the buccal vein of the mice every 4 weeks after feeding for analysis. Then, the CD-and HFD-fed mice were each further divided into tumor-bearing or tumor-free mice (n = 10 per each group). In the tumor-bearing mice, approximately 4 × 10^6^ MC38 CRC cells were subcutaneously inoculated on the middle of the backs of mice to analyze tumor progression. Three weeks after tumor cell inoculation, the mice were sacrificed, and the immune cells of the tumor, blood, and various organs were analyzed. The tumors were measured using calipers. Tumor volume was calculated using the following formula:(1)Tumor volume =m1×m2×m2×0.5236,
where m1 is the length of the longer axis and m2 is the length of the shorter axis. Similarly, for survival analysis, CD- and HFD-fed male mice were inoculated with approximately 4 × 10^6^ MC38 CRC cell line and analyzed for 60 days (n = 10 per each group). After 60 days, all mice were euthanized via cervical dislocation under anesthesia.

### 2.4. Antibody-Mediated Depletion of CD4+ T Cells in Mice

To investigate how CD4+ T cells contribute to tumor growth suppression, a separate group of mice fed the HFD or CD (n = 7 per each group) for 9 weeks were administered 150 μg/mouse anti-mouse CD4a antibody (clone GK1.5; BioXCell Lebanon, NH, USA) or anti-mouse CD8a antibody (clone 53-5.8; BioXCell Lebanon, NH, USA) intraperitoneally every three days starting from the day before MC38 cell inoculation. Mice administered rat immunoglobulin (IgG)2b (clone LTF-2; BioXCell Lebanon, NH, USA) served as the control group for this experiment.

### 2.5. Fluorescence-Activated cell Sorting (FACS)

Blood samples were collected from the buccal vein and inferior vena cava of mice after sacrifice in heparinized tubes using a 23G needle (Terumo, Tokyo, Japan). Blood samples were diluted with phosphate-buffered saline (PBS) containing 0.1% bovine serum albumin (BSA) and layered on Histopaque 1119 to isolate murine peripheral blood mononuclear cells (PBMCs) and centrifuged (500*× g*, 25 °C, 20 min). The spleens and lymph nodes were harvested from mice, respectively, crushed, and passed through a cell strainer (BD Biosciences) of a nylon mesh with 70-μm pores for isolation of lymphocytes. The tumors were cut into small sections and digested in 1 mg/mL collagenase III (Worthington) and 60 U/mL DNase I (Roche) solution at 37 °C for 60 min. The cells were then passed through a cell strainer with 40-μm pores and centrifuged at 500*× g* for 5 min. Following discarding the supernatant, the cells were resuspended in 5 mL of 40% Percoll (GE Healthcare), centrifuged at 500× *g* for 10 min. All samples were incubated at 37 °C for 10 min in hemolysis buffer containing 139.5 mM NH_4_Cl and 1.7 mM Tris-HCl (pH 7.65) and washed with 0.1% BSA/PBS. Mouse Fc-blocker (Miltenyi Biotec, Bergisch Gladbach, Germany) was treated to block nonspecific binding sites, and incubated with the antibody mixture (Appendix A) at 4 °C for 20 min, as described previously [20]. All antibodies were purchased from BioLegend (San Diego, CA, USA). Flow cytometry was performed on a FACSVerse instrument (BD Biosciences), and data were analyzed using FlowJo software (TreeStar, Inc.).

### 2.6. Measurement of Intracellular Cytokine Production

To measure intracellular cytokine production, 1.0 × 10^6^ cells/mL lymphocytes isolated from MC38 tumor cells and splenocytes were incubated with a leukocyte activation cocktail (Cat#; 550583, BD Biosciences Pharmingen) for 6 h at 37 °C. Cell surface antigens were stained by incubation with mixed antibodies (Appendix A) at 4 °C for 20 min. Cells were fixed and permeabilized at 4 °C for 20 min in the dark using a cytofix/cytoperm kit (Cat#; 554714, BD Biosciences Pharmingen). After washing with washing solution (BD Perm/Wash™ Buffer), cytokines were stained with antibodies (Appendix A).

### 2.7. Immunohistochemistry Staining

Tumor tissue samples were obtained 3 weeks after inoculating CRC cell line MC38, and the specimens were fixed in formalin. Paraffin-embedded tissue was sliced into 5 μM-thick slices, and sections were soaked and deparaffinized in xylene and ethanol with a concentration gradient for immunofluorescence. Blocking of nonspecific binding was performed by incubating 10% bovine serum albumin (BSA) at 25 ℃ for 1 h. Slides were incubated overnight at 4 ℃ with the following antibodies (1:100, diluted with 10% BSA/PBS): anti-CD8a mAb(Cat#: 98941, CST, MA, USA) and anti-CD4a mAb(Cat#: 25229, CST, MA, USA) and anti-PD-1 mAb(Cat#: 84651, CST, MA, USA) and anti-CD107a mAb(Cat#: 46843, CST, MA, USA). Following two washes with 2% BSA/PBS, samples were stained with 0.1 μg/mL DAPI (blue) for DNA staining and incubated for 20 min at 25 ℃ in the dark. The samples were then imaged using a KEYENCE BZ-710 microscope (Keyence Corporation, Osaka, Japan).

### 2.8. Statistical Analysis

Differences between the two groups were analyzed using Student’s *t*-test if normally distributed; otherwise, using Mann–Whitney’s non-parametric test was used. One-way or two-way analysis of variance with Holm’s post hoc test was performed to analyze differences among multiple groups. When analyzing the effect and the interaction of two factors, two-way analysis of variance (2 × 2) was used. Continuous variables are expressed as means. Survival curves were generated using the Kaplan–Meier method and analyzed using the log-rank test. All *P* values < 0.05 were considered indicative of a statistically significant difference. All statistical analyses were performed using EZR [21] (Saitama Medical Center, Jichi Medical University, Saitama, Japan), a graphical user interface for R (The R Foundation for Statistical Computing, Vienna, Austria). EZR is a modified version of the R commander designed to add functions frequently used in biostatistics.

## 3. Results

### 3.1. HFD-Induced Obesity Reduces CD4+ T Cell Population in Peripheral Blood and Gut-Associated Lymphoid Tissues (GALTs)

After 9 weeks of feeding (Figure 1a), the HFD-fed mice had a significantly increased body weight compared to CD-fed mice (Figure 1b). Flow cytometry results revealed that the number and frequency of NK cells in the blood were significantly reduced in HFD-fed mice compared to those in CD-fed mice at 8 weeks (Appendix A). Moreover, the number and frequency of CD4+ T cells in peripheral blood significantly reduced in HFD-fed mice compared to those in CD-fed mice at 4 weeks (Figure 1c); however, reduction in CD8+ T cell number was not observed in HFD-fed mice. Furthermore, the expression of programmed death-1 (PD-1) in CD4+ and CD8+ T cells in HFD-fed mice was significantly higher than that in CD-fed mice (Figure 1d). In the thymus and several secondary lymphoid tissues, no reduction in CD8+ T cell number was observed in HFD-fed mice, whereas a significant reduction in CD4+ T cell number was observed in GALTs, such as mesenteric lymph nodes and Peyer’s patches, at 12 weeks (Figure 1e, Appendix A). These results show that HFD feeding causes a CD4+ T cell-specific reduction in peripheral blood and GALTs.

### 3.2. HFD-Induced Obesity Accelerates Tumor Growth and Impairs Survival

Three weeks after tumor inoculation (Figure 2a), in the CD group, no change in body weight or epididymal white adipose tissue (eWAT) weight was observed in tumor-bearing or tumor-free mice. In the HFD group, tumor-bearing mice showed significant body weight loss and decreased eWAT weight compared to those of tumor-free mice (Figure 2b,c). HFD feeding also accelerated tumor growth and impaired the survival rate (Figure 2d,e).

### 3.3. Tumor Burden Further Promotes Reduction in the Number and Exhaustion of CD4+ T Cells in the Blood of HFD-Induced Obese Mice

Three weeks after inoculating with MC38 cells, the CD4+ T cell number in the blood of HFD-fed mice was significantly reduced compared to that in CD-fed mice (Figure 3a). HFD-fed mice showed a marked reduction in CD4+ T cell number in the blood after tumor inoculation; no such difference was observed in the CD-fed mice. The same patterns were observed for CD8+ T cells (Figure 3b). FACS analysis revealed no differences in the proportions of naïve (CD44^low^ CD62L^high^), effector memory (CD44^high^ CD62L^low^), and central memory (CD44^high^ CD62L^high^) subpopulations of CD4+ and CD8+ T cells in the blood between CD- and HFD-fed tumor-bearing mice (Figure 3c). In contrast, PD-1 expression, especially the percentage of PD-1^high^ cells, which are functionally impaired in many cases [22], of CD4+ and CD8+ T cells, was significantly increased in the HFD-fed tumor-bearing mice (Figure 3d).

### 3.4. HFD-Induced Obesity Substantially Promotes Reduction in the Number and Dysfunction of CD4+ T Cells in Tumors

Similar to how tumor inoculation promoted the HFD-induced reduction in the number of CD4+ T cells in the blood, the number of CD4+ T cells in the tumors was significantly reduced in HFD-fed mice compared to that in CD-fed mice (55.9 vs. 18.2 cells/mg; mean difference 37.7 cells/mg; 95% CI 14.9–60.3 cells/mg; *p = 0.006*). Similarly, the number of CD8+ T cells in the tumors was reduced. This trend was more pronounced for CD4+ T cells than for CD8+ T cells (Figure 4a). In the phenotypical analysis, CD4+ T cells in tumors of HFD-fed mice showed a shift from the naïve to effector memory phenotype (Figure 4b), which was not observed in CD8+ T cells. PD-1 expression in tumors, especially the percentage of PD-1^high^ cells, was significantly increased in both CD4+ and CD8+ T cells in HFD-fed mice (PD-1^high^ of CD4+; 23.3% vs. 39.0%; mean difference −15.7%; 95% CI −39.0%–−4.9% *p = 0.009*) (Figure 4c). In fluorescent immunostaining assays, a marked reduction in the number of CD4+ T cells and an increase in PD-1+ CD4+ T cells in HFD-fed mice compared to those in CD-fed mice (Figure 4d) were observed. This finding was more pronounced in the tumor center. We examined the expression levels of CD107a, a degranulation marker that shows cytotoxic activity against tumor cells in CD4+ [23] and CD8+ T cells [24]. The expression level of CD107a of CD4+ and CD8+ T cells in the tumors of HFD-fed mice significantly reduced (CD107a on CD4+; 26.6% vs. 15.2%; mean difference 11.4%; 95% CI 4.1–18.7 % *p = 0.006*) (Figure 5a). In fluorescent immunostaining, CD107a+ CD4+ T cells were barely detectable in the tumor centers of HFD-fed mice (Figure 5b). Associated with this, their ability to produce cytokines was evaluated to characterize the functional properties of CD4+ and CD8+ T cells. The production of cytokines such as IFN-γ and TNF-α was significantly reduced in CD4+ T cells and CD8+ T cells of HFD-fed mice (Figure 5c). In contrast, there were no significant differences in granzyme B. In CD4+ T cells, there were no significant differences in IL-2 and IL-21 expression (Appendix A). We examined whether the increased PD-1^high^ cells of CD4+ T cells in the tumors of HFD-fed mice was associated with reduced cytokine production. In CD4+ T cells, the expression of TNF-α was significantly higher in PD-1^neg^ cells and IFN-γ expression was higher in PD-1^int^ cells, but those in HFD-fed mice were lower than those in CD-fed mice. Furthermore, the PD-1^high^ cells expressed lower quantities of TNF-α and IFN-γ than other cells (Figure 5d). These results show that HFD-induced obesity leads to dysfunctional status resulting from the exhaustion and decreased cytotoxic activity of CD4+ T cells in tumors.

### 3.5. HFD-Induced Obesity Accelerates Tumor Growth in a CD4+ T Cell-Dependent Manner

To determine whether the acceleration of tumor growth in HFD-fed mice is involved in the reduction of CD4+ T cells, we performed a depletion analysis according to a previous report [11]. Depletion of CD4+ T cells in the blood following antibody treatment was confirmed (Figure 6a, Appendix A), followed by subcutaneous tumor inoculation. In CD4+ T cell-depleted mice, acceleration of tumor growth was observed in CD-fed mice but not in HFD-fed mice (Figure 6b). The feeding-dependent acceleration of tumor growth during control IgG administration was not observed in mice depleted of CD4+ T cells (Figure 6c). The same results were observed for CD8+ T cells (Figure 6c). These results indicate that the acceleration of tumor growth due to HFD-induced obesity compared to control is not only CD8+ T cell-dependent but is also CD4+ T cell-dependent. We next analyzed the effects of CD4+ T cells on CD8+ T cells in each organ in both CD- and HFD-fed mice. CD4+ T cell depletion markedly reduced the number of CD8+ T cells in the tumor, as well as the effect of obesity, with no such reduction being observed in the blood and spleen (Figure 6d). Furthermore, CD4+ T cell depletion similarly upregulated PD-1 expression in the remaining CD8+ T cells in the tumor in both CD- and HFD-fed mice (Figure 6e). This indicates that the number and PD-1 expression of CD8+ T cells in the tumors are affected not only by obesity but also by CD4+ T cells. The feeding-dependent decrease in CD107a expression of CD8+ T cells during control IgG administration was not observed in mice depleted of CD4+ T cells (Figure 6f). This suggests that CD107a expression of CD8+ T cells, which is reduced in HFD-fed mice, was caused by the reduced number and dysfunction of CD4+ cells due to HFD-induced obesity.

## 4. Discussion

In this study, we showed the reduced number and anti-tumor immune dysfunction of CD4+ T cells in the obesity-associated TIME of MC38 cells. First, we hypothesized that immune dysfunction of CD4+ T cells accelerates tumor growth due to HFD-induced obesity. To address this hypothesis, we established an obesity mouse model (fed 45% HFD). We found that in HFD-fed mice, the CD4+ T cells in the peripheral blood were reduced in number and more exhausted over time with the development of obesity. HFD-induced obesity accelerated MC38 tumor growth and impaired survival, similar to the results of previous reports [11]. Furthermore, in the present study, HFD-induced obesity promoted the reduction in number and exhaustion of CD4+ T cells in the peripheral blood by tumor inoculation and similarly led to the reduction in number and exhaustion of CD8+ and CD4+ T cells in the tumor. When CD4+ T cells were depleted in vivo, tumor growth was accelerated in CD-fed mice compared to that in the IgG group. When comparing the CD4+ T cell-depleted groups, no significant differences in tumor growth were observed between diets. Based on these findings, we concluded that the reduced number and anti-tumor immune dysfunction of CD4+ T cells accelerated tumor growth due to HFD-induced obesity.

We clearly showed the reduction in CD4+ T cell number in the peripheral blood and GALTs in HFD-fed mice prior to tumor inoculation. Previous reports have shown that HFD feeding reduces the number of naïve CD4+ and CD8+ T cells because of thymus atrophy [25] and impairs intestinal immunity [26]. We assumed that a similar mechanism could be attributed to our results; however, abnormal T cell maturation in the thymus and reduction in CD8+ T cell number in the blood or GALTs were not observed. The discrepancies between these reports and our results could be attributed to the difference in the composition of HFD (our study: 45% HFD, previous study: 60% HFD). Thus, our results strongly suggest that CD4+ T cells are more susceptible to lipotoxicity than CD8+ T cells. The only report on the selective reduction in CD4+ T cell numbers supports our results [27]. This previous report showed that in nonalcoholic fatty liver disease, intrahepatic CD4+ T cells in humans and mice are selectively impaired by the accumulation of mitochondria-derived ROS induced by high volumes of fatty acids, which contributes to hepatocellular carcinoma progression. We are currently focusing on the reactivity of CD4+ T cells to some fatty acids and the cause of CD4+ T cell reduction.

In the present study, CD4+ and CD8+ T cells in the tumors of HFD-fed mice showed increased expression of PD-1 and decreased expression of CD107a. In addition, CD4+ T cells of HFD-fed mice showed impaired production of cytokines such as IFN-γ and TNF-α. These results indicate that the direct anti-tumor effects of CD4+ T cells of HFD-fed mice are impaired both phenotypically and functionally. We also evaluated the association between the degree of PD-1 expression and functional characteristics. The relation between PD-1 expression and functions such as cytokine production in CD4+ T cells is not as well understood as that in CD8+ T cells [28]. The impact of obesity on the exhaustion of CD4+ T cells, including PD-1 expression, requires further investigation. In terms of direct anti-tumor immune responses, little is known about CD4+ T cells, though recent studies have demonstrated that CD4+ T cells also produce granzymes and perforins with cytotoxic activity as an anti-tumor effect [13,14]. In the obesity-associated TIME, elevated fatty acid levels inhibit glycolysis in effector CD8+ T cells through the leptin/PD-1-STAT3 pathway and reduce anti-tumor activity [29]. Patsoukis et al. [30] have shown that PD-1 signaling promotes fatty acid β-oxidation and inhibits glycolysis in human CD4+ T cells. A similar mechanism can be assumed for CD4+ T cells in obesity because our results are compatible with these reports.

The present study is a validation using a model under limited conditions in which a single cell line is inoculated into mice with diet-induced obesity. In terms of colorectal cancer, to clarify the TIME and tumor progression, we need to investigate an orthotropic model [31] or a genetically engineered mouse model [32]. Furthermore, the obesity model is even more complicated. The mouse model is limited because it only captures one aspect of the pathology of obesity [33]. There are still only a few mouse models of 45% HFD, and furthermore, these models require standardization. Other varieties of mouse models differ not only in body weight, fat mass, and metabolic factors but also in inflammation and related factors [34,35,36]. These may also cause significant differences in tumor progression and the TIME, and the choice of model should be based on the pathophysiology. We are conducting a detailed study of 45% and 60% HFD. We wish to build on this and use genetically engineered mouse models to gain further insights into obesity and the progression of colorectal cancer.

Our model revealed that the acceleration of tumor growth due to obesity is not only CD8+ T cell-dependent, as previously reported [11], but is also CD4+ T cell-dependent. These findings support the critical role of CD4+ T cells in the obesity-associated TIME and that the reduced anti-tumor activity of CD4+ T cells contributes to the acceleration of tumor growth mediated through an effect on CD8+ T cells. Indeed, our CD4-depletion study results showed that the number and exhaustion of CD8+ T cells in tumors were affected by both CD4-depletion and HFD-induced obesity, respectively. Our results also suggest that, in the obesity-associated TIME, the reduced cytotoxic activity of CD8+ T cells was caused by CD4+ T cell dysfunction. Furthermore, many reports showed a direct effect of obesity on impaired cytotoxic activity of CD8+ T cells in obesity-associated TIME [11,37,38]. Few reports have reported their association with CD4+ T cells [39]. Regarding the association between CD8+ and CD4+ T cells, CD4+ T cells are considered to function as “helpers” of CD8+ T cells. Previous reports have shown that CD4+ T cells are essential for the optimal induction of anti-tumor responses by CD8+ T cell adoptive transfer and are involved in regulating PD-1 expression [40]. A more recent report also identified an essential role for CD4+ T cells in promoting the differentiation of superior cytotoxic effector CD8+ T cells [41]. Collectively, these findings suggest that decreased “helper” function due to CD4+ T cell reduction and exhaustion in obesity-associated TIME led to decreased cytotoxic activity of CD8+ T cells.

## 5. Conclusions

In conclusion, we showed that the reduction and anti-tumor immune dysfunction of CD4+ T cells contribute to the acceleration of tumor growth in the obesity-associated TIME in tumor-bearing 45% HFD-fed mice. Our findings of the impact of HFD-induced obesity on the anti-tumor activity of CD4+ T cells may provide a clue to elucidate the pathogenesis of poor outcomes of CRC with obesity comorbidities.

## Figures and Tables

**Figure 1 cells-12-00086-f001:**
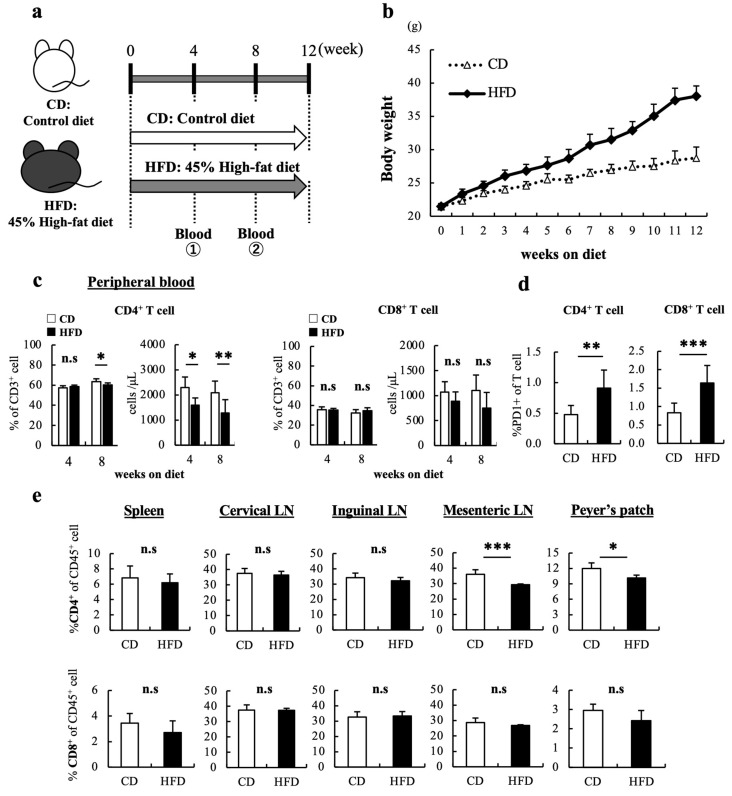
High-fat diet (HFD)-induced obesity reduced CD4+ cells in the peripheral blood and gut-associated lymph nodes of mice. (**a**) Experimental schema is shown. Male mice were fed a HFD (45% calories from fat) or CD (10% calories from fat) for 12 weeks. (**b**) The change in body weight of mice fed the CD and those fed the HFD was measured. (**c**) Blood sample was obtained from the buccal vein, and changes over time in the number and percentage of T cells in the peripheral blood were analyzed at 4 and 8 weeks. (**d**) PD-1 expression in T cells in peripheral blood at 8 weeks are shown. (**e**) Percentage of T cell in CD45+ cells in secondary lymphoid tissues at 12 weeks are shown. The data are expressed as the mean ± standard deviation (S.D.) ((**a**–**d**), n = 20 mice in each group; (**e**), n = 10 mice in each group). Student’s *t*-test was performed for two-group comparisons. * *p* < 0.05, ** *p* < 0.01, *** *p* < 0.005. HFD, High-fat diet; CD, Control diet.

**Figure 2 cells-12-00086-f002:**
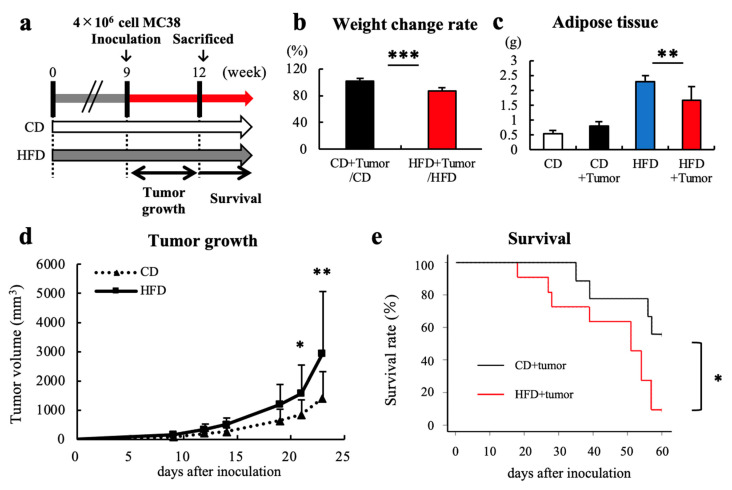
HFD-induced obesity accelerated tumor growth and impaired survival. (**a**) The experimental schema is shown. After 9 weeks of feeding, mice were further divided into tumor-bearing and tumor-free groups (n = 10). For the tumor-bearing group, the colon cancer cell line MC38 (4 × 10^6^ cells) was inoculated subcutaneously on the backs of the mice. (**b**) After 3 weeks of tumor inoculation, the weight ratio of tumor-bearing to tumor-free mice in each group was measured. (**c**) The weight of eWAT in each group were measured. (**d**) Tumor volume and **(e**) survival rate during 3 weeks of tumor inoculation were measured. Mann–Whitney’s test was performed for two-group comparisons. Survival curves were drawn using the Kaplan–Meier method and analyzed using the log-rank test. * *p* < 0.05, ** *p* < 0.01, *** *p* < 0.005. HFD, High-fat diet; CD, Control diet; eWAT, epididymal adipose tissue.

**Figure 3 cells-12-00086-f003:**
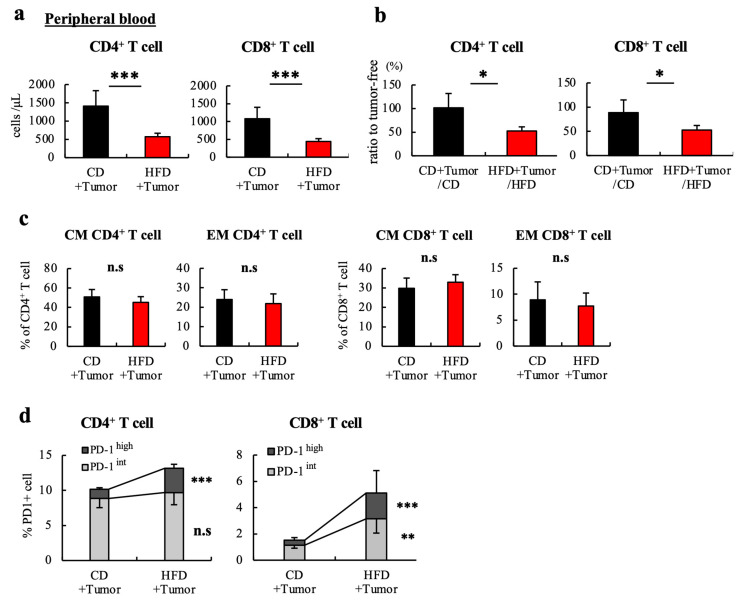
Tumor burden further promotes reduction in the number and exhaustion of CD4+ T cells in the blood of HFD-induced obese mice. (**a**) Three weeks after tumor inoculation, the number of T cells in the peripheral blood of each group were analyzed. (**b**) Ratio of T cell numbers in peripheral blood with tumor-bearing to tumor-free mice in each group are shown. (**c**) The phenotype of peripheral blood T cells of tumor-bearing mice at 12 weeks are shown. The CM and EM of T cells were defined as CD44+ CD62L+ and CD44+ CD62L- cells, respectively. (**d**) The PD-1 expression of peripheral blood T cells of tumor-bearing mice at 12 weeks are shown. PD-1 expression of T cells was analyzed in three groups. The data are expressed as the mean ± S.D. (n = 10 mice in each group). Mann–Whitney’s test was performed for two-group comparisons. * *p* < 0.05, ** *p* < 0.01, *** *p* < 0.005. HFD, High-fat diet; CD, Control diet; CM, central memory; EM, effector memory; PD-1, programmed death-1.

**Figure 4 cells-12-00086-f004:**
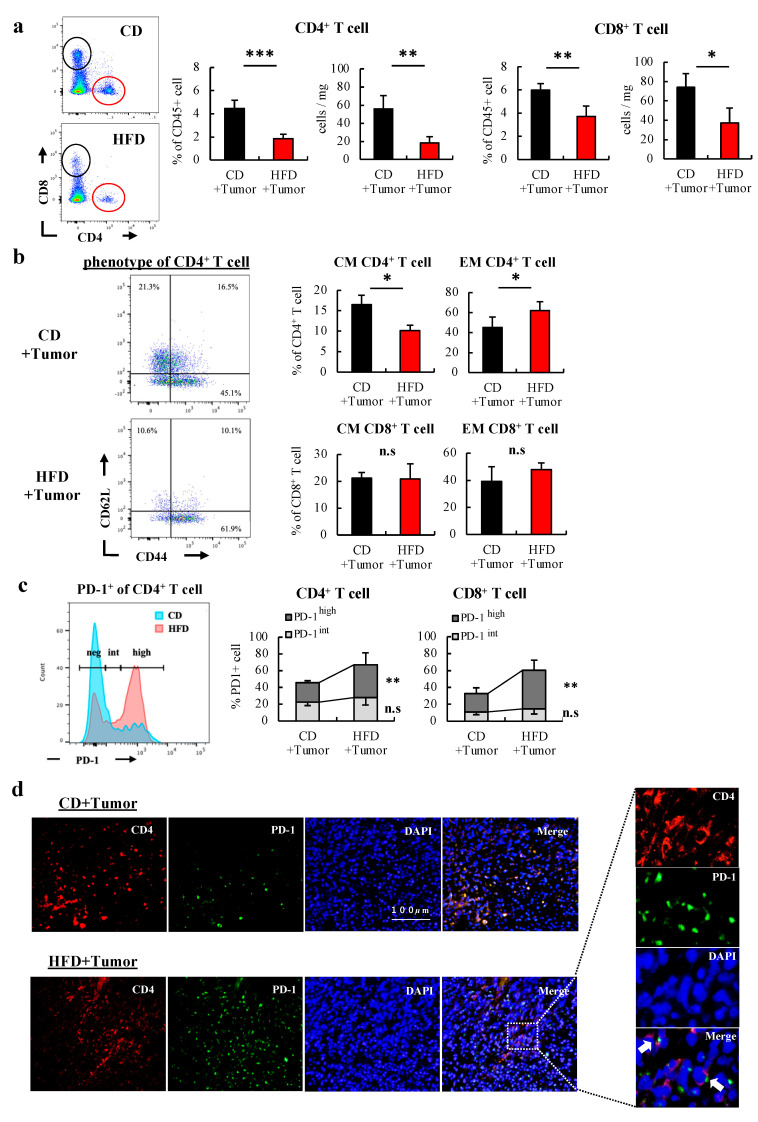
HFD-induced obesity substantially promoted the reduction in number and PD-1 expression of CD4+ T cells in tumors. The CD4+ T cells in the tumors of CD-and HFD-fed mice were analyzed after 3 weeks of tumor inoculation. (**a**) The percentage of CD4+ and CD8+ T cells in CD45+ cells in the tumor and the cell number per mg of tumor were analyzed using flow cytometry. (**b**) The phenotype of T cells in the tumors are shown. A representative figure of phenotype of CD4+ T cell in tumors is shown. The CM and EM of T cells were defined as CD44+ CD62L+ and CD44+ CD62L- cells, respectively. (**c**) A representative histogram of PD-1 expression of CD4+ T cell in tumors is shown. PD-1 expression of T cells is analyzed in three groups. (**d**) Representative immunofluorescent images of MC38 tumor sections from CD- and HFD-fed mice; CD4+ T cells (red), PD-1+ cells (green), DAPI (blue), white arrows (PD-1+ CD4+ T cell). The data are expressed as the mean ± S.D. (n = 10 mice in each group). Mann–Whitney’s test was performed for two-group comparisons. * *p* < 0.05, ** *p* < 0.01, *** *p* < 0.005. HFD, High-fat diet; CD, Control diet; CM, central memory; EM, effector memory; PD-1, programmed death-1; neg, negative; int, intermediate.

**Figure 5 cells-12-00086-f005:**
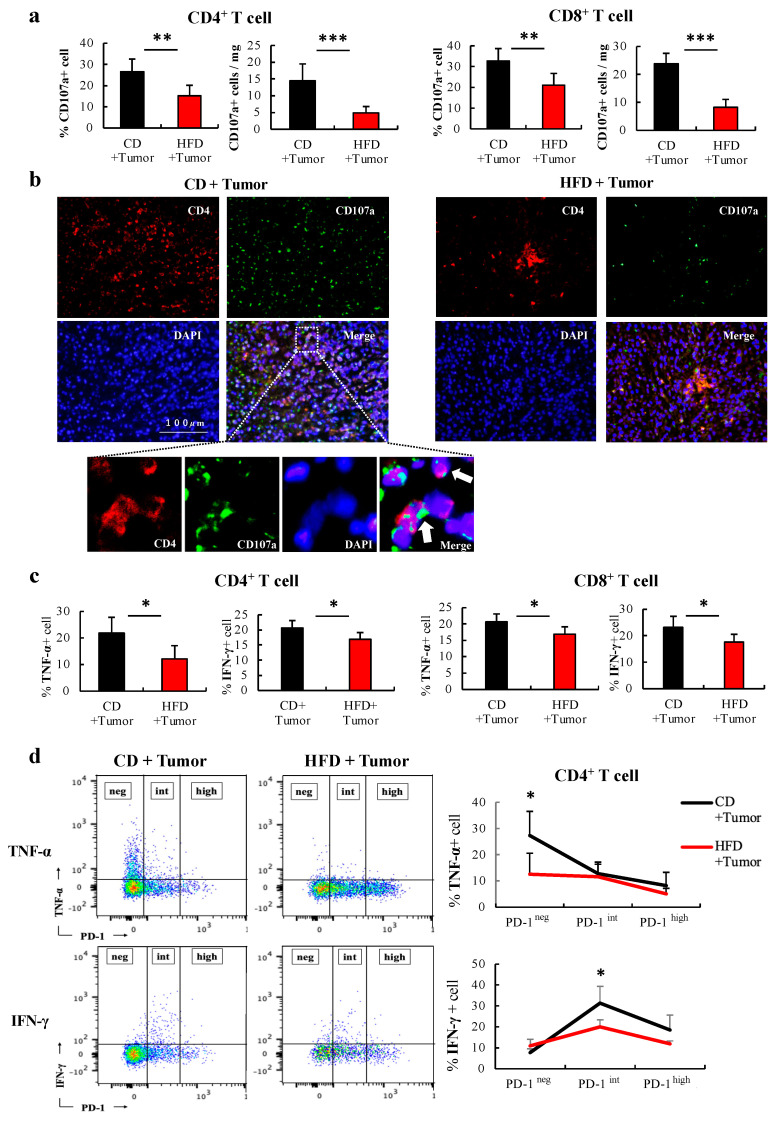
HFD-induced obesity attenuates the cytotoxic activity of CD4+ T cells in tumors. The CD4+ T cells in the tumors of CD-and HFD-fed mice were analyzed after 3 weeks of tumor inoculation. (**a**) The percentage and number of CD107a+ expression of T cells in the tumor are shown. (**b)** Representative immunofluorescent images of MC38 tumor sections from CD- and HFD-fed mice; CD4+ T cells (red), CD107a+ cells (green), DAPI (blue), white arrows (CD107a+ CD4+ T cell). (**c**) The percentage of TNF-α+ expression and IFN-γ of T cells in the tumor following stimulation with PMA+ ionomycin for 6 h are shown. (**d**) Representative flow cytometry and percentage of cytokine IFN-γ and TNF-α expression in PD-1^neg^, PD-1^int^, and PD-1^high^ CD4+ T cells, respectively. The data are expressed as the mean ± S.D. (n = 5 mice in each group). Mann–Whitney’s test was performed for two-group comparisons. * *p* < 0.05, ** *p* < 0.01, *** *p* < 0.005. HFD, High-fat diet; CD, Control diet; PD-1, programmed death-1; PMA, Phorbol12-myristate13-acetate; neg, negative; int, intermediate.

**Figure 6 cells-12-00086-f006:**
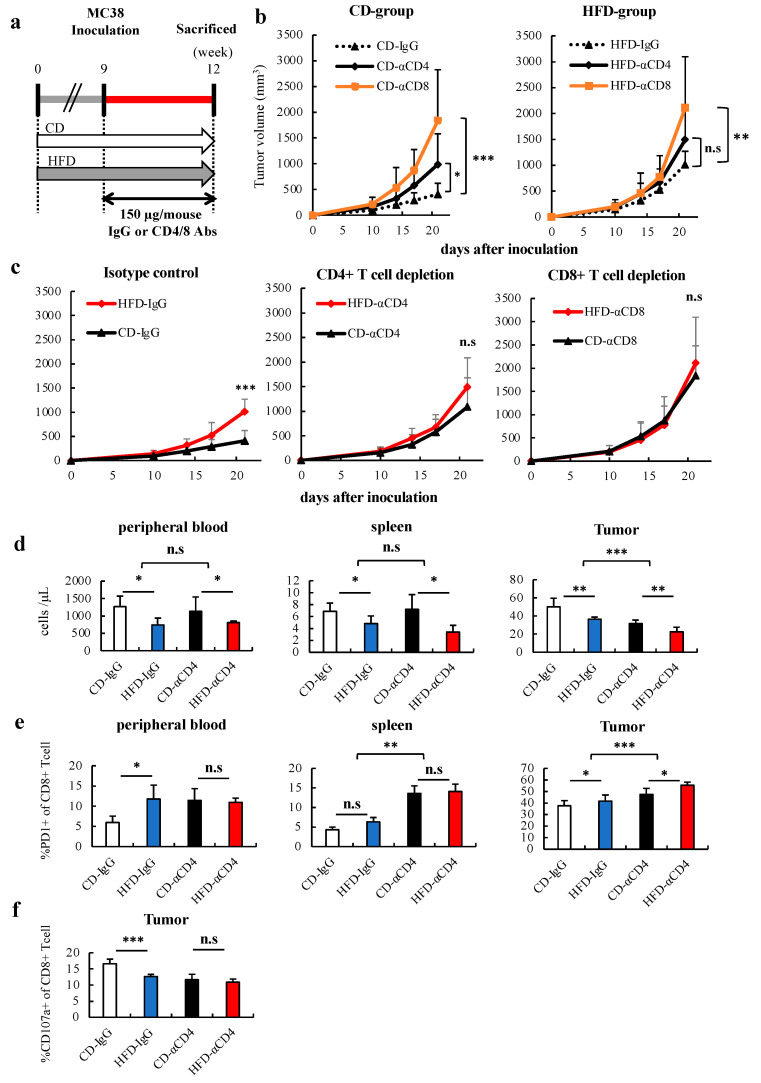
HFD-induced obesity accelerated tumor growth in a CD4+ T cell-dependent manner. (**a**) Experimental schema is shown. After 9 weeks of feeding, the colon cancer cell line MC38 (4 × 10^6^ cells) was inoculated subcutaneously on the backs of mice. Anti-CD4 antibodies were administered starting from the day before tumor inoculation, and the effect on tumor progression was observed. (**b**) Tumor volume of the CD- and HFD-fed groups during 3 weeks of tumor inoculation was measured. (**c**) Tumor volume of the CD4- and CD8-depleted and isotype groups during 3 weeks of tumor inoculation was measured. (**d**) the number and (**e**) PD-1 expression of CD8+ T cells in each organ were analyzed. (**f**) The percentage of CD107a expression in CD8 + T cells in tumors following CD4 depletion are shown. The data are expressed as the mean ± S.D. (n = 7 mice in each group). Two-way analysis of variance (2 × 2) was used to analyze the effects and the interaction of two factors: diet (unrelated to anti-CD4) or anti-CD4 (unrelated to diet). * *p* < 0.05, ** *p* < 0.01, *** *p* < 0.005. HFD, High-fat diet; CD, Control diet; PD-1, programmed death-1.

## Data Availability

The datasets generated and/or analyzed during the current study are available from the corresponding author on reasonable request.

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
