# Peer review of "Reduced Number and Immune Dysfunction of CD4+ T Cells in Obesity Accelerate Colorectal Cancer Progression"

_cells, 2022, doi:10.3390/cells12010086_

Round 1

Reviewer 1 Report

Here authors report how High fat diet affects CD4+ and CD8+ T cell number and function in vivo, that can further influence tumor growth. Using MC38 mouse CRC cell line injected into HFD and CD fed mice, they show that CD4 and CD8 T cell number is depleted in blood and in the tumor microenvironment. They also show that there is an increase in PD1 high cells in HFD fed mice compared to CD fed mice.  Finally they confirm these finding by injecting anti-CD4 and anti-CD8 antibodies into tumor bearing mice on CD or HFD that resulted in further increase in tumor size. This data is convincing but not very surprising as many studies have report HFD induced Obesity as a risk factor for cancer prognosis and therapeutic resistance via multiple mechanisms including immune modulation. 

There are few points that needs to be addressed .....

Major

1. When HFD leads to depleted CD4 and CD8 cells that results in increased tumor size/poor survival (Figs 1-4), the secondary approach to confirm this findings would be to adaptive transfer of function CD4 / CD8 cells. But surprisingly authors choose to deplete the CD4/CD8 cells further using antibodies. Explain why?

2. Ln 226. Authors say the hypothesis was that "CD4 T cells accelerate tumor growth due HFD induced obesity". But the data indicates that CD4 T cells are depleted and existing ones are non-functional/PD1 high. These two are contradicting, Please clarify and correct the sentence.

3.Ln 235. "CD4 depletion abolished tumor growth by HFD feeding". This is not correct. Suppl. Fig 1e clearly shows antibody treatment increased the tumor size compared to controls in both CD and HFD groups. Please clarify and correct the sentence.

4. Fig 5b comparing CD vs HFD is not the objective of this expt. Rather tumor growth between control vs antibody treated mice should be shown here. So, Fig 5b should be replaced with suppl. Fig 1e.  Similarly, all the other figs in Fig 5 the comparison should be between control and antibody treated groups (within CD or HFD). 

Minor

5. Abstract Ln 25, should include the MC38 word for clarity

6. Intro. Ln 42-44 is not clear. please reframe.

7. Methods Section 2.3, include gender of mice used. Also include volume of injections for MC38 cells.

Author Response

Response to Reviewer #1:

Major

Comment 1: When HFD leads to depleted CD4 and CD8 cells that results in increased tumor size/poor survival (Figs 1-4), the secondary approach to confirm this findings would be to adaptive transfer of function CD4 / CD8 cells. But surprisingly authors choose to deplete the CD4/CD8 cells further using antibodies. Explain why?

Response 1: Thank you for your critical question. In our previous preliminary experiment, we performed the adoptive transfer analysis of CD4+ T cells derived from CD-fed mice into HFD-fed mice, but this could not suppress the progression of tumor growth in HFD-fed mice. The reason may be that CD4+ T cells transferred into the HFD host are functionally impaired and consequently less likely to exert their anti-tumor effects. We have shown that the number of CD4+ T cells in the peripheral blood reduced 4 weeks after the feeding of HFD (Figure 1c), thus suggesting that the transferred CD4+ T cells could be affected relatively early, within 4 weeks. Furthermore, given the tumor effect, it was difficult to fully complement the antitumor effect with a transfer analysis.

According to the reviewer’s comment, we revised the corresponding statements as follows for greater clarity (Line 248):

To determine whether the acceleration of tumor growth in HFD-fed mice is involved in the reduction of CD4+ T cells, we performed a depletion analysis according to a previous report [11].

Comment 2: Ln 226. Authors say the hypothesis was that "CD4 T cells accelerate tumor growth due HFD induced obesity". But the data indicates that CD4 T cells are depleted and existing ones are non-functional/PD1 high. These two are contradicting, Please clarify and correct the sentence.

Response 2: According to the reviewer’s comment, we changed the text in the Discussion section of the revised manuscript as follows (Line 271):

First, we hypothesized that immune dysfunction of CD4+ T cells accelerates tumor growth due to HFD-induced obesity.

Comment 3: Ln 235. "CD4 depletion abolished tumor growth by HFD feeding". This is not correct. Suppl. Fig 1e clearly shows antibody treatment increased the tumor size compared to controls in both CD and HFD groups. Please clarify and correct the sentence.

Response 3: The original wording of the manuscript was misleading. According to the reviewer’s suggestion, we changed the text in the Discussion as follows (Line 280):

When CD4+ T cells were depleted in vivo, tumor growth was accelerated in CD-fed mice compared to that in the IgG group. When comparing the CD4+ T cell-depleted groups, no significant differences in tumor growth were observed between diets.

Comment 4: Fig 5b comparing CD vs HFD is not the objective of this expt. Rather tumor growth between control vs antibody treated mice should be shown here. So, Fig 5b should be replaced with suppl. Fig 1e.  Similarly, all the other figs in Fig 5 the comparison should be between control and antibody treated groups (within CD or HFD). 

Response 4: We partially agree with the reviewer's comments. Supplementary Fig 1e was changed to Figure 6b, and the original Figure 5b was also retained as Figure 6c for consistency with the original discussion. In addition, we revised the corresponding statements as follows for greater clarity (Line 251): 

In CD4+ T cell-depleted mice, acceleration of tumor growth was observed in CD-fed mice but not in HFD-fed mice (Figure 6b). The feeding-dependent acceleration of tumor growth during control IgG administration was not observed in mice depleted of CD4+ T cells (Figure 6c). The same results were observed for CD8+ T cells (Figure 6c). These results indicate that the acceleration of tumor growth due to HFD-induced obesity compared to control is not only CD8+ T cell-dependent but is also CD4+ T cell-dependent.

Minor

Comment 5: Abstract Ln 25, should include the MC38 word for clarity

Response 5: As per the reviewer’s comment, we changed the text in the Abstract as follows: (Line 25):

A tumor-bearing obese mouse model was established by feeding with 45% high-fat diet (HFD), followed by inoculation with a colon cancer cell line MC38.

Comment 6: Intro. Ln 42-44 is not clear. please reframe.

Ln 42-44 (Because the global prevalence of obesity nearly tripled between 1975 and 2016, with 39% of adults being overweight and obesity prevalent in 13% of the worldwide population [6], the number of patients with CRC is estimated to further increase.)

Response 6: As per the reviewer’s comment, we changed the text in the Introduction as follows (Line 42):

The global prevalence of obesity nearly tripled between 1975 and 2016, with 39% of adults being overweight (body mass index (BMI)25 kg/m2) and 13% being obese (BMI30 kg/m2) worldwide [6]. The latest research suggests that the risk of CRC correlates with increasing BMI [7]. Furthermore, it is estimated that the number of patients with CRC will further increase and the number of new cases and deaths will exceed 2.2 million and 1.1 million, respectively, by 2030 [2].

Comment 7: Methods Section 2.3, include gender of mice used. Also include volume of injections for MC38 cells.

Response 7: As per the reviewer’s comment, we included information regarding the gender of the mice and the volume of injections for MC38 cells in the Methods subsection 2.3.

Reviewer 2 Report

Interesting, elegant and well planned study.

Given the small sample size, it would be interesting to reflect the confidence intervals for more refined conclusions

Author Response

Response to Reviewer #2:

Comment 1: Interesting, elegant and well planned study.

Given the small sample size, it would be interesting to reflect the confidence intervals for more refined conclusions

Response 1: Thank you for your important point. Since it would be complicated to reflect confidence intervals in all analyses, confidence intervals were incorporated for important data.

As per the reviewer’s comment, we changed the text in the Results section as follows (Line 213):

Similar to how tumor inoculation promoted the HFD-induced reduction in the number of CD4+ T cells in the blood, the number of CD4+ T cells in the tumors was significantly reduced in HFD-fed mice compared to that in CD-fed mice (55.9 vs 18.2 cells/mg; mean difference 37.7 cells/mg; 95% CI 14.9–60.3 cells/mg; P = 0.006).

PD-1 expression in tumors, especially the percentage of PD-1high cells, was significantly increased in both CD4+ and CD8+ T cells in HFD-fed mice (PD-1high of CD4+; 23.3% vs 39.0%; mean difference -15.7%; 95% CI -39.0%– -4.9% P = 0.009) (Figure 4c).  

The expression level of CD107a of CD4+ and CD8+ T cells in the tumors of HFD-fed mice significantly reduced (CD107a on CD4+; 26.6% vs 15.2%; mean difference 11.4%; 95% CI 4.1–18.7 % P = 0.006) (Figure 5a).  

Reviewer 3 Report

This manuscript described the impact of obesity on CD4+ T cells in tumour immune microenvironment in colorectal cancer. Overall, the manuscript was well written in good English and met the high quality standard of the journal. The study design is elegant, the method used is appropriate, the results are well presented and clearly described, and the data interpretation is proper. It is an important study that showed depletion of CD4+ T cells in obese model reduced tumour infiltration, increased PD-1 expression and accelerated the tumour growth.

I list below my comments for improvement.

1. This is an elegant in vivo study. However, the colorectal cancer model was solely relied on syngeneic subcutaneous graft in which may not be the best model to mimic the actual colorectal cancer. There has been increasing evidence that the immune microenvironment in subcutaneous tissue is significantly different from the gastrointestinal tract. Perhaps it will be best to address this limitation in the discussion.

2. Supplementary Data, Figure c: Some Chinese characters partially blocked the graphs. Please rectify.

3. Supplementary Figure Legend 1(a): Authors wrote “..changes over time in the number and percentage of T cells in the peripheral blood were analyzed at 4.8 weeks”?? Please confirm the duration.

Author Response

Response to Reviewer #3:

Response 1: As per the reviewer’s comment, we added the following statements to the manuscript for greater clarity (Line 318):

The present study is a validation using a model under limited conditions in which a single cell line is inoculated into mice with diet-induced obesity. In terms of colorectal cancer, to clarify the TIME and tumor progression, we need to investigate an orthotropic model [31] or a genetically engineered mouse model [32].

Comment 2: Supplementary Data, Figure c: Some Chinese characters partially blocked the graphs. Please rectify.

Response 2: We have corrected the formatting error the reviewer pointed out in Supplementary Data, Figure 1c.

Comment 3: Supplementary Figure Legend 1(a): Authors wrote “..changes over time in the number and percentage of T cells in the peripheral blood were analyzed at 4.8 weeks”?? Please confirm the duration.

Response 3: We corrected the error in the legend for Figure 1c and Supplementary Figure 1a as per the reviewer’s suggestion.

figure Legend 1(c)..changes over time in the number and percentage of T cells in the peripheral blood were analyzed at 4 and 8 weeks

Supplementary Figure Legend 1(a).. changes over time in the number and percentage of NK cells in the peripheral blood were analyzed at 4 and 8 weeks.